# Beta Diversity of Arbuscular Mycorrhizal Communities Increases in Time after Crop Establishment of Peruvian Sacha Inchi (*Plukenetia volubilis*)

**DOI:** 10.3390/jof9020194

**Published:** 2023-02-02

**Authors:** Ana Maria de la Sota Ricaldi, Sofía Rengifo del Águila, Raúl Blas Sevillano, Álvaro López-García, Mike Anderson Corazon-Guivin

**Affiliations:** 1Laboratorio de Biología y Genética Molecular, Universidad Nacional de San Martín, Jr. Amorarca N° 315, Morales 22201, Peru; 2Facultad de Agronomía, Universidad Nacional Agraria la Molina, Av. La Molina s/n, Lima 15024, Peru; 3Instituto Interuniversitario de Investigación del Sistema Tierra en Andalucía (IISTA), Universidad de Jaén, 23071 Jaén, Spain; 4Department of Soil Microbiology and Symbiotic Systems, Estación Experimental del Zaidín (EEZ-CSIC), 18008 Granada, Spain

**Keywords:** *Plukenetia volubilis*, arbuscular mycorrhizal fungi, agroecosystem, T-RFLP, community assembly

## Abstract

(1) Background: Beta diversity, i.e., the variance in species compositions across communities, has been pointed out as a main factor for explaining ecosystem functioning. However, few studies have directly tested the effect of crop establishment on beta diversity. We studied beta diversity patterns of arbuscular mycorrhizal (AM) fungal communities associated to sacha inchi (*Plukenetia volubilis*) after crop establishment. (2) Methods: We molecularly characterized the AM fungal communities associated to roots of sacha inchi in plots after different times of crop establishment, from less than one year to older than three. We analyzed the patterns of alpha, beta, and phylogenetic diversity, and the sources of variation of AM fungal community composition. (3) Results: Beta diversity increased in the older plots, but no temporal effect in alpha or phylogenetic diversity was found. The AM fungal community composition was driven by environmental factors (altitude and soil conditions). A part of this variation could be attributed to differences between sampled locations (expressed as geographic coordinates). Crop age, in turn, affected the composition with no interactions with the environmental conditions or spatial location. (4) Conclusions: These results point out towards a certain recovery of the soil microbiota after sacha inchi establishment. This fact could be attributed to the low-impact management associated to this tropical crop.

## 1. Introduction

Sacha inchi (*Plukenetia volubilis* L.), is an Amazon evergreen liana species cultivated in Peru and in different tropical regions of the world [1,2,3,4]. It contains edible seeds of high nutritional value because they contain 45–50% of essential fatty acids (composed of approximately 77.5–84.4% polyunsaturated fatty acids, 8.4–13.2% monounsaturated fatty acids, and 6.8–9.1% saturated fatty acids) and 22–30% of proteins, tocopherols, phytosterols, and antioxidants, characteristics that have given a commercial importance to this species, mostly for the pharmaceutical industry [1,4,5,6,7,8]. In Peru, it is traditionally cultivated mostly as a monoculture. It can also be found as part of agroforestry systems or associated with subsistence crops or other annual or perennial crops such as coffee or cocoa [9,10].

The culture of sacha inchi is usually established after subsistence crops such as *Zea mays* or *Musa* sp. in secondary forests or on abandoned lands [11,12]. For this, as a first step, the removal of the previous vegetation layer is usually done by slash–and-burn [13]. After this practice, the soil starts to impoverish by the alteration of the microbiome, the physical and chemical properties of the soil, and/or soil erosion, which, in time, impedes the maintenance of other crops and finally leads to the abandonment of land [14,15,16]. As a keystone for the soil functionality and provision of ecosystem services, the change in the soil microbiome must have a main role in this process [17,18].

A key group of soil microbiome is the arbuscular mycorrhizal (AM) fungi. These fungi establish mutualistic associations with a wide variety of plants [19]. This association allows the plant to improve the uptake of mineral nutrients from the soil (mainly, nitrogen and phosphorus) and low-moving ions (phosphate, ammonia, zinc, copper) through the fungal hyphal system [20,21,22]. It also promotes water balance and helps protect plants from soil pathogens while providing an increased tolerance to abiotic stresses (salt, water) and heavy metals in moderately contaminated soils [23,24,25]. In response to the foregoing, the fungus is given carbon compounds derived from photosynthesis [26].

At present, few studies have been done on the interaction of these fungi with *Plukenetia volubilis* species and their diversity. Previous studies under controlled conditions (greenhouse) have shown that AM fungi (genera: *Paraglomus*, *Glomus*, *Acaulospora*, *Claroideoglomus*, and *Funneliformis*) stimulate growth and development and improve the drought tolerance of sacha inchi seedlings through alterations in the morphological, physiological, and biochemical characteristics of the roots [27,28]. Likewise, some AM fungal species have been reported in sacha inchi rhizospheric soils in the Peruvian rainforest and in Thailand [29,30,31,32,33]. However, these reports correspond to the taxonomic classification of spores; the diversity and composition of AM fungi colonizing sacha inchi roots is still unknown.

The diversity, spore density, and community composition of AM fungi are often influenced by different environmental factors [34,35,36]. In agroecosystems, one of the most important factors is the agricultural management that is thought to cause a filter in soil microbiomes, selecting for microorganisms adapted to anthropogenic modified conditions [37]. Such conditioning is assumed to induce temporal changes in the community composition of crop-associated microbes [38], reducing the spatial variability (i.e., beta diversity) present in a particular site and, hence, indirectly reducing the resilience of the system [39]. Indeed, although beta diversity has been pointed out as a main factor for the explaining ecosystem functioning [40], few studies have directly tested the effect of crop establishment on beta diversity.

Here, we aimed to study the beta diversity of AM fungal communities associated to sacha inchi in two different moments after crop establishment. We hypothesized a higher variability between AM fungal communities, i.e., higher beta diversity, immediately after crop establishment due to the combined effect of the inherent spatial variability in the original tropical forest soils and the chaotic nature of the disturbance caused by the land-use conversion. By contrast, the homogeneity generated by the monocropping will cause a homogenization across AM fungal communities in time. In addition, we also aimed to describe the main AM fungal taxonomic diversity associated to sacha inchi and the main environmental drivers of the assembly of their community composition: geography, soil or topographic variables, and age.

## 2. Materials and Methods

### 2.1. Study Place

The study was carried out in four localities in the San Martín region (Figure 1), an area located in the northern jungle of the Peruvian territory between 305 and 900 m.a.s.l. (Table A1). The average temperature in these localities ranges between 26 and 27.5 °C (minimum annual average ranges from 20 to 22 °C, while maximum ranges from 32 to 33 °C), while annual rainfall is between 1502 and 2680 mm [41]. These conditions allow for the growth of various cultivable species, such as sacha inchi. This crop is usually established on lands that have been previously cultivated with crops such as cocoa (*Theobroma cacao*) or coffee (*Coffea arabica*) in areas where the soil has lost its productive capacity. In addition, due to its rusticity, sacha inchi plantations are managed in traditional ways and are commonly associated with other crops such as banana (*Musa* sp.), papaya (*Carica papaya*), and guaba (*Inga feuilleei*).

### 2.2. Sampling Design

From each locality, two plots were selected: one with plantations less than one year old, and the other older than three years old. In each plot, four plants were randomly selected and spaced by at least by 15 m, while three subsamples (ca. 7 g each) of roots per plant were carefully collected by tracing from the stem up to 15 cm apart and placed on ice. In addition, root-associated soil was collected at a depth between 5 and 25 cm from each sub-sampled point and mixed into a composite sample for physico-chemical analysis. Altitude of sampled plots was also annotated as explanatory variable.

### 2.3. Soil Analysis

The soil samples (500 g) were air-dried for 48 h and sieved through 2 mm mesh. The pH was measured using a potentiometer in a 1:1 soil:water suspension. Salinity was measured as the electrical conductivity of the aqueous extract in a soil:water 1:1 ratio. The available phosphorus content was determined by the modified Olsen method (extraction with 0.5M NaHCO_3_ pH 8.5) [42]. Available potassium was measured by extraction with ammonium acetate (pH 7.0). The calcium carbonate content was calculated by the gasovolumetric method using a calcimeter. Organic matter was determined using the Walkley and Black method by oxidation of organic carbon with potassium dichromate (%OM = %C × 1.724) [43]. The texture was determined using the hydrometer method [44]. Cation exchange capacity was determined by saturation with ammonium acetate (pH 7.0) [45]. Exchangeable cations were determined by replacing with ammonium acetate (pH 7.0) quantified by flame photometry [45]. The micro-Kjeldahl method was used to determine the percentage of nitrogen [46]. The analysis was carried out in the Laboratorio de Análisis de Suelos, Plantas, Aguas y Fertilizantes of the Agronomy Faculty of the Universidad Nacional Agraria la Molina (LASPAF-UNALM). A summary of soil characteristics per study site can be found in Table A2.

### 2.4. Root Molecular Analysis

Roots were washed with distilled water and dried with absorbent paper, then cut into small segments (ca. 0.5 cm). An aliquot (100 mg fresh weight) of the finest and best-preserved fragments were stored at −80 °C for further processing. DNA extraction was made using the CTAB method [47] after crushing roots in a mortar with liquid nitrogen.

A fingerprinting of the AM fungal community composition was arranged by terminal restriction fragment length poly morphism (T-RFLP). For that, a gene library containing the genetic diversity of AM fungi contained in the whole study was built, following the known T-RFLP database approach [48,49]. A nested PCR was performed to amplify a partial region of the large sub-unit (LSU) of the rDNA of AM fungi. The first reaction was performed using the set of primers SSUmAr and LSUmAf [50], 0.02 U/μL of Taq KOD, 1× of buffer for KOD DNA polymerase, 1.5 mM of magnesium chloride, 0.4 mM of dNTPs, and 0.375 ng/μL of DNA for a final volume of 10 μL. The PCR conditions were initial denaturation of 95 °C for 5 min, 40 cycles of 95 °C for 30 s, 58 °C for 45 s, and 75 °C for 2 min, followed by a final elongation at 72 °C for 10 min. The second reaction was performed using the primer set eukaryotic FLR3 [51] and specific for AMF NDL22 [52], 0.5 μL of the initial PCR product, 0.02 U/μL of Taq PLATINUM polymerase, 1× of Taq Hi Fi PLATINUM buffer, 2 mM of magnesium sulfate, 0.2 mM of dNTPs, for a final volume of 25 μL. PCR conditions were initial denaturation at 94 °C for 2 min, 33 cycles of 94 °C for 30 s, 62 °C for 30 s, and 68 °C for 40 s with a final elongation of 68 °C for 10 min. The presence of the fragments of interest was verified by agarose gel electrophoresis. The reaction was run separately for every sample and an equimolar mixture of all PCR products was prepared and purified using the GFX PCR DNA and gel band purification kit (GE Healthcare Life Sciences, Freiburg, Germany). This mixture was ligated into the PCR 2.1 vector and cloned using chemically competent cells of *E. coli* One Shot TOP10 (Invitrogen, Carlsbad, CA, USA) by heat shock. A total of 71 clones was sequenced using the sequencing service of Macrogen Inc. Korea and the M13 primers set. Sequences are available at NCBI GenBank under accession numbers OQ132701-OQ132763.

The sequences obtained were aligned using MAFFT version 7 [53], and the primer sequences were removed. Due to the known genetic variability inside AM fungal clades, we used the monophyletic clade approach to define operational taxonomic units (OTUs) [54]. Thus, in a first step, sequences were clustered at 97% similarity using the Mothur software [55]. Secondly, representative sequences of each 97% cluster were subjected to a phylogenetic network construction using SplitsTree version 4.14.8 [56]. In the phylogenetic network, sequences in the same terminal clade without reticulation were considered as belonging to the same OTU (Figure A1). Subsequently, a blast against GenBank was arranged to name and taxonomically assign the found OTUs. As a way to represent the phylogenetic position across known AM fungal species, a phylogenetic tree was also performed by grouping the representative sequences of each OTU with sequences of other AM fungal species available in the GenBank database (NBCI) and the fungus Boletus edulis (AF336240) as outgroup. For this, a neighbour-joining algorithm with the Kimura-2 parameter as a substitution model and 1000 replications as a bootstrap value was implemented in MEGA X version 10.0.5 [57]. Once OTUs were defined, the diversity coverage of the clone library was tested by a rarefaction analysis using vegan R package.

After the gene library was completed, a set of four restriction enzymes able to discriminate across the found AM fungal OTUs was selected using the online software REPK version 1.3 [58]: Hpy8I, BsedI, AanI, and TaiI. Theoretical terminal fragment sizes (TRFs) were determined by using TriFle [59]. Using the PCR products of the first reaction as templates, two nested PCRs were prepare using the primers FLR3 (marked either with fluorescent 6-FAM or with PET) and NDL22. The products of these PCR were digested separately with the selected enzymes. PCR products marked with 6-FAM were digested separately with 5 U of the enzymes Hpy8I and BsedI, and those marked with PET were digested with 5 U of the enzymes AanI and TaiI. Incubation temperatures were 37 °C, 55 °C, 37 °C, and 65 °C for 16 h, respectively. Subsequently, the digested products belonging to the same sample were mixed as follows: Hpy8I marked with FAM with AanI marked with PET; and BsedI marked with FAM with TaiI marked with PET. The purification of the products was carried out using the QIAEX II Gel Extraction purification kit (QIAGEN), following the manufacturer’s instructions. The TRF profiles were analyzed at Macrogen Korea Inc. through the Genescan service in multiplex. Empirical TRF sizes for the cloned genes of the gene library were obtained by applying the same methodology. The TRF profiles of samples and clones were processed using Peak Scanner version 2.0 (Applied Biosystem). To match the sample TRF profiles with empirical TRF sizes of the gene library, we used the TRAMPR package [60] with a threshold level of 3 bp, excluding fragments smaller than 65 and larger than 460 bp. The data obtained from the matching were organized as a matrix of OTU presence/absence per sample.

On the other hand, an additional analysis was made using the T-RFLP peak profile approach [48], the profiles of the enzymes BsedI and TaiI (those showing the maximal fragment variation), and excluding fragments smaller than 65 bp and larger than 460 bp and peaks present in a single sample and those that contributed to less than 1% to the total peak area [61]. In this way, an abundance matrix based on peak areas was prepared to allow quantitative comparisons across TRF profiles.

### 2.5. Statistical Analyses

Richness of OTUs in each sample was calculated as the number of OTUs reported. Coverage of the sampling was assessed by calculating species accumulation curves and species coverage (SC) by each studied site and age using the iNEXT R package [62], considering incidence frequency counts. Beta diversity was determined as the dissimilarity across fungal communities using the Jaccard distance for the database T-RFLP approach and the Bray–Curtis dissimilarity for the Peak-profile approach (vegdist function, vegan R package).

To measure the phylogenetic structure of the AM fungal community, we calculated the standardized effect size of mean phylogenetic pairwise distance (SES–MPD) using picante R package [63]. For this calculation, the OTU × OTU phylogenetic distance matrix was obtained with MEGA X from the previously generated phylogenetic tree. To obtain the SES-MPD, 999 randomized null communities were generated using the independent swap algorithm. Subsequently, the mean value of SES–MPD by age range was used to judge the significance of phylogenetic clustering or segregation of the AM fungal communities. The statistical significance of the SES–MPD for each level of analyzed factors was tested using a *t*-test.

We used linear mixed models to test the effect of age (fixed factor), including the sampled site as random factor on OTU richness, beta diversity, and SES-MPD. Age of sampled sites was coded as a two-levels factor (Range A, 0–1 y, vs. Range B, 3.5–6 y), and response variables were logarithmically transformed to improve model fitting.

To study the drivers of AM fungal community composition, the explained variance of their communities was partitioned into soil variables, site age, and spatial distribution [64]. The peak-profile table was Hellinger-transformed for this analysis. Log-transformed environmental variables were previously selected to keep the least number of most explanatory variables by forward selection of variables (ordistep function, vegan R package). Spatial distribution of AM fungal communities was analyzed by including the spatial position (geographic coordinates) in the analysis. Hence, differences across sites were included in the analysis. Age factor was, in this case, introduced as a continuous variable. The significance of individual partitions was tested via partial redundancy analysis. Finally, to visualize the found patterns, an RDA ordination was performed constraining the AM fungal community variation to the selected variables used in the variation partitioning analysis.

All analyses were approached with R version 3.5.3 (R Core Team, 2019).

## 3. Results

### 3.1. Characterization of AM Fungal Community

The rarefaction analysis of the gene library predicted a total of 16 OTUs from which 13 were included in the library (ca. 81% of the total, Figure A2). The 13 OTUs belonged to three different families: Diversisporaceae, Acaulosporaceae, and Glomeraceae, with one, one, and seven OTUs, respectively (Table A3). Most of the identified OTUs did not correspond to known Glomeromycotan species. Only one OTU was identified as the species *Acaulospora scrobiculata*, with an identity percentage of 98.91%. The species accumulation analysis revealed a good sampling coverage ranging from 100% to 88%, per the studied site and age (Figure A3).

### 3.2. Effect of Age on Diversity of AM Fungal Communities

Linear mixed models showed that age had a positive significant effect on beta diversity when analyzed using the database approach (Table 1, Figure 2). No effect was detected for OTU richness or phylogenetic diversity (SES–MPD). In the case of the peak profile approach, the linear mixed models only showed the locality influence in beta diversity.

### 3.3. Drivers of Community Assembly (Database)

The partition of variation showed that environmental factors alone explained 6% of the total variance of the AMF communities (F_df_ = 1.590_3,25_, *p* = 0.037), age explained 3% (F_df_ = 1.913_1,25_, *p* = 0.039), and the covariance between the environment and space explained 5% of the variation in AM fungal communities (Figure 3). Spatial position alone had no effect. The selected environmental factors (altitude, the total sum of bases, and the CaCO_3_ content in the soil) were those best for explaining the variation of AM fungal communities (Figure 3). Likewise, the RDA model, including these variables together with age and location, explained a total variance of 12.9% (*p* = 0.001).

### 3.4. Drivers of Community Assembly (Peak Profile)

For the peak profile approach, the partition of variation showed that the spatial position alone explained 6% of the total variance of the AM fungal communities (F_df_ = 2.050_1,26_, *p* = 0.006), age explained 5% (F_df_ = 2.616_2,26_, *p* = 0.008), environmental factors explained 4% (F_df_ = 1.617_2,26_, *p* = 0.034), and its covariance with space explained 7% (Figure 4). The selected environmental factors were soil organic matter and soil-available phosphorus. Likewise, the RDA model explained 13.5% of the total variance (*p* = 0.001).

## 4. Discussion

### 4.1. Variability of Beta Diversity and Composition of AM Fungal Communities

Our results indicated no influence of any factor (locality or age) on AM fungal richness (i.e., number of OTUs) or the phylogenetic diversity of their communities. However, the variability across AM fungal communities in the studied plots (beta diversity) was influenced by the age of sacha inchi plantations. Contrary to our expectations and previous results in tropical systems [65,66], we found an increase of beta diversity with time. AM fungal community compositions in roots of plants less than one-year-old were more homogeneous, while those older than three years showed to be significantly more heterogeneous, i.e., they shared a lower number of species. Other previous studies have reported a decrease in the diversity of AM fungi over time. For example, in the case of the cultivable species *Panax ginseng* or the tree species *Tetragastris panamensis* and rubber (*Hevea brasiliensis*) in tropical systems [39,40]. A possible explanation for this result could be a shift in the AM fungal preferences along the plant development. Thus, in young plantations, AM fungi with a greater capacity for colonization and dispersing, i.e. those with ruderal life-history traits, would predominate [67]. The greater dispersal of this group would downgrade the differences across sampled plants in plots. Meanwhile, in adult plantations, those AM fungi with greater persistence capacity, probably more symbiotically active, better competitor fungi *sensu* Chagnon et al. [67], would have replaced the first AM fungal ruderal colonizers.

This result could be also related to the sacha inchi culture management in Peru. Usually, this species is established when the soil has lost its productive capacity [9,11,12]. This previous management could have caused the found homogenizing effect immediately after sacha inchi establishment [68], and the low impacting sacha inchi managements could have contributed to increase the beta diversity in time. The plantations of this species are managed with little agronomic work and little human intervention, since they are usually managed during the first and third year after sowing. The diversity of AM fungi is affected by a number of factors, such as anthropogenic activity, environmental conditions, and stochastic processes [34,35,36,37,38,39]. Among the most studied are mechanical agricultural practices, which cause a major disturbance [69,70,71]. Likewise, the use of fertilizers has a reducing effect on mycorrhizal colonization [72,73]. However, these factors do not affect AM fungal taxa in the same manner because some families show tolerance, attributed mainly to their life cycle [67]. The very low use of these practices in the sacha inchi cultivation could be behind the lack of a homogenizing effect due to anthropogenic effects.

### 4.2. Influence of Soil Physico-Chemical Variables on the Composition of AM Fungal Communities

This change in beta diversity was linked to a compositional turnover of AM fungi in time and was due to environmental variables. We used two different approaches to study the shift of AM fungal community composition: T-RFLP database and peak profile approaches, as they allowed to study different aspects of mycorrhizal community ecology [48,49].

The database approach showed a main influence of environmental variables and the age of the plantation on AM fungal community composition. According to the RDA, the environmental variables that significantly influenced the composition of the community of AM fungi were the content of calcium carbonates in the soil (CaCO_3_), the sum of bases, and the altitude (m.a.s.l.). Although, as far as we know, previous studies have not found any relationship between CaCO_3_ and the composition of AM fungi, some authors report that high levels of CaCO_3_ have some influence on the growth of extraradical hyphae and spore germination and also reduce the number of arbuscules [74]. On the other hand, this variable is directly related to changes in soil pH. The soils studied showed a direct correlation between pH and the concentration of CaCO_3_ [75,76]. It is possible that this variable is indirectly showing the effect of pH on AM fungi, which is known to affect the activity of AM fungi and alter the composition of the AM fungi community [77,78].

Furthermore, the sum of bases corresponding to the sum of the cations Ca, Mg, K, and Na (meq/100 g) also drove the composition of AM fungi. Previous studies have shown the influence of the changeable cations Ca, Mg, and K on AM fungal spore community variation [79]. However, the mechanisms that determine this effect have not been investigated. Despite this, these elements are involved in the established symbiosis between AM fungi and plants, since moderate levels of these cations are involved in the stimulation of root colonization. Ca promotes the permeability of root cells, so its action would be related to the establishment of symbiosis [80]. K is considered a necessary element for maintaining the symbiosis and promotes mycorrhizal colonization through the ability to carry this element of AM fungi to plants roots [81]. Moreover, these fungi allow it to maintain the K/Na balance, which contributes to the selective intake of ions and, therefore, the improvement of plant nutrition [82,83]. On the other hand, Mg is an essential element of chlorophyll, so its deficiency mainly affects the synthesis and, therefore, the reduction of carbon necessary to establish the symbiosis [81]). In this sense, in general, these cations improve nutrient availability conditions, which helps to improve nutrient imbalances in the leaves so that the plant will make a greater photosynthate production and stimulate mycorrhizal associations [84].

Regarding the peak profile approach, the results showed that the behavior of the AM fungal community was mainly influenced by environmental variables, spatial location, and the plantation age. The selected environmental variables were the available P and the soil organic matter content. The amount of P in soil is negatively related to the colonization and richness of AM fungal species [85,86]. It also causes changes in the composition of communities [87]. P in soil is an immobilized element appearing as organic P, CaPO_4,_ or other fixed forms [23,87]. AM fungi improved soil PO_4_ acquisition for plants by exploring the soil to places where the plant is unable to reach [80]. In response, the plant could identify and reward the most beneficial partners for it. For this reason, it would be expected to find a greater diversity of AM fungi interacting with the plant in an environment where the nutrient availability is limited [85,88].

On the other hand, changes in organic matter composition may also affect the growth and composition of AM fungi since this improves soil fertility, activating a more diverse group of microorganisms [73]. The predominance of some species of AM fungi has been linked to the type of amendments used as fertilizer [73,89,90]. Furthermore, they showed changes in the abundance of certain species of the genus *Glomus* in response to the application of high amounts of organic matter, while the abundance of species of the genus *Paraglomus* was favored by the application of low doses.

## Figures and Tables

**Figure 1 jof-09-00194-f001:**
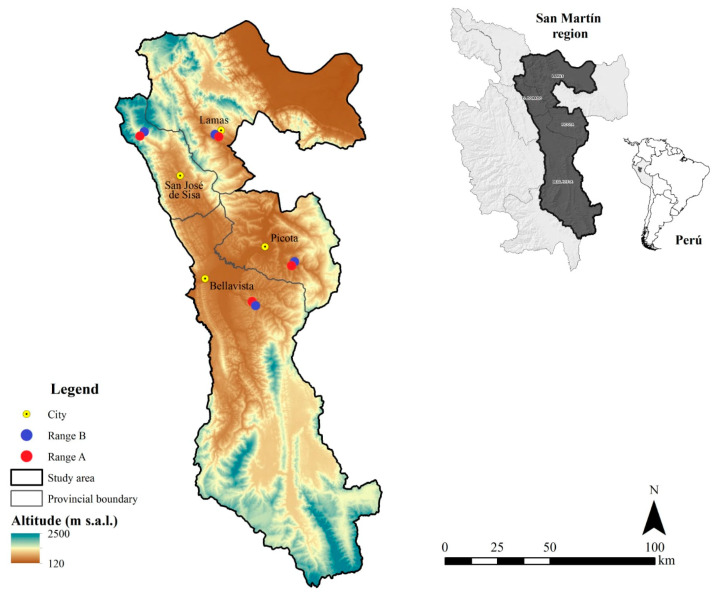
Spatial location of the eight sampled sacha inchi production plots, located in four localities of the San Martín region in Peru. Red points represent plots with plantations less than one year old, and blue points represent those older than three years old.

**Figure 2 jof-09-00194-f002:**
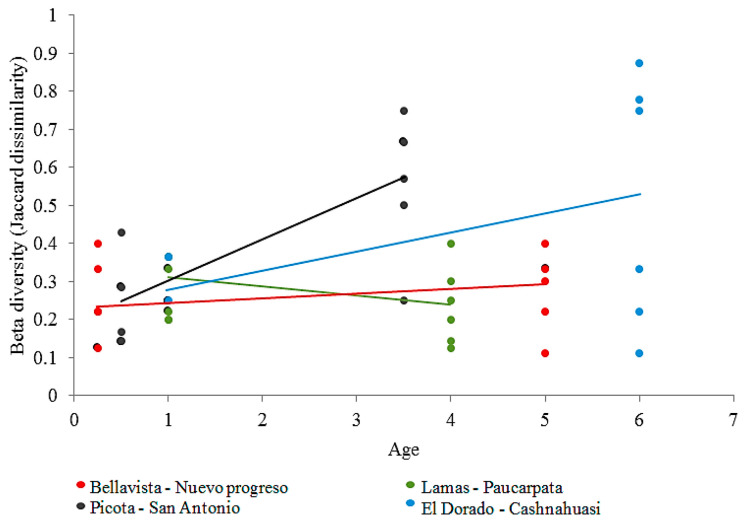
Variation of beta diversity (Jaccard dissimilarity across sample pairs, *y*-axis) of AM fungal communities across crop ages (years, in *x*-axis) and localities (different color).

**Figure 3 jof-09-00194-f003:**
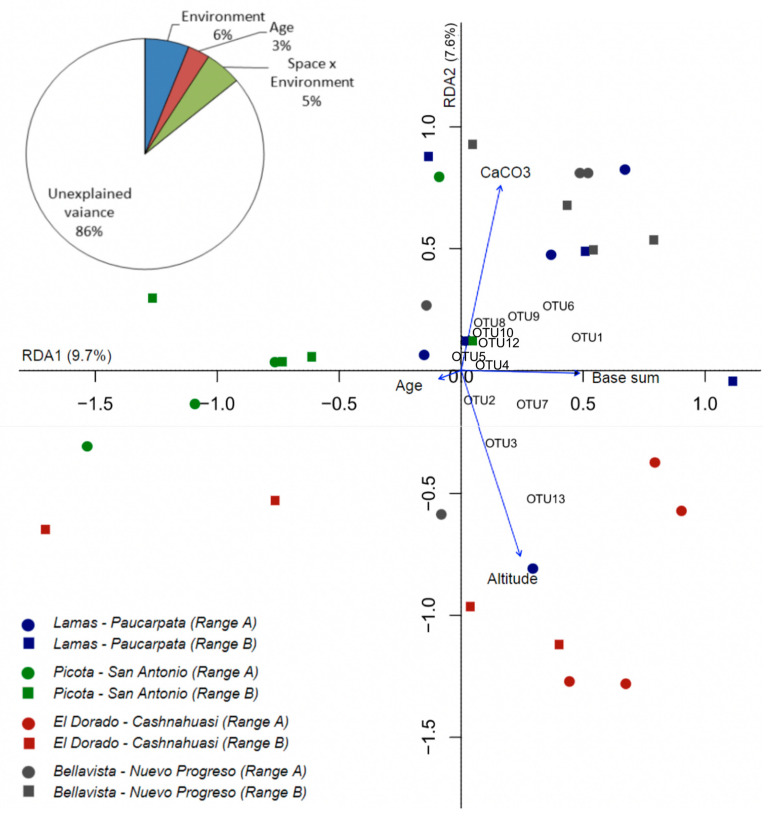
Ordination of the arbuscular mycorrhizal fungi community composition via redundancy analysis (RDA) (TRFLP database approach) in four localities in the San Martín region. The position of samples in the figure indicates their similarity (closer samples, more similar community composition) and correlation with shown variables (samples in the direction of arrows contains AM fungal taxa whose presence correlates with those variables). The pie chart shows a variance partitioning of the community composition by environmental factors (CaCO_3_, base sum and altitude), plot spatial position, and crop age. In the figure, age, altitude, CaCO_3,_ and base sum soil content were used as constraints.

**Figure 4 jof-09-00194-f004:**
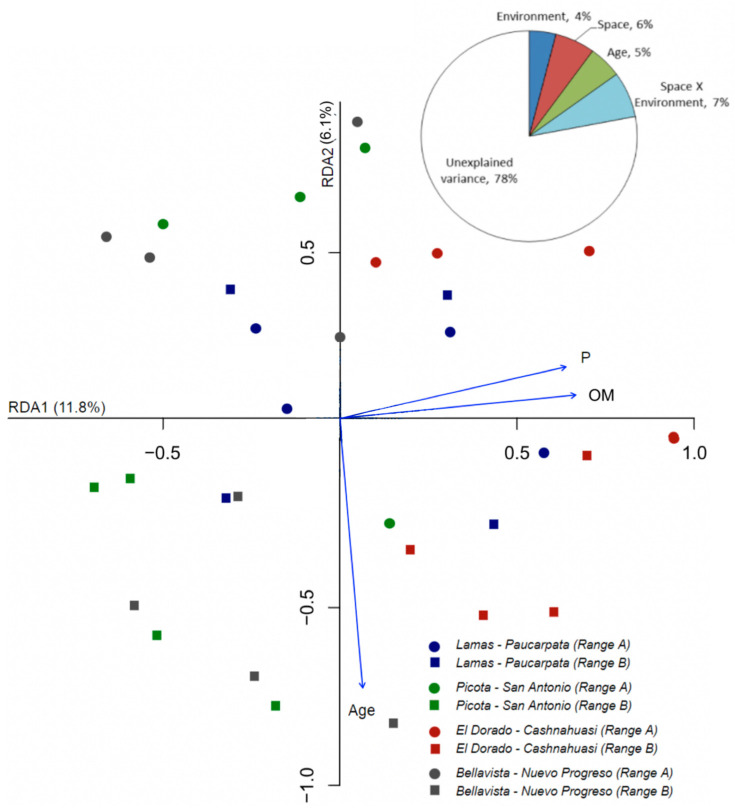
Ordination of the arbuscular mycorrhizal fungi community composition via redundancy analysis (RDA) (TRFLP peak profile approach) in four localities in the San Martín region. The pie chart shows a variance partition of the community composition explained by environmental factors (soil organic matter -OM- and P), plot spatial position, and crop age. In the figure, age, soil OM, and soil P content were used as constraints.

**Table 1 jof-09-00194-t001:** Results of linear mixed models testing the effect of age on arbuscular mycorrhizal diversity metrics for both TRFLP database and peak profile approaches. F values (Likelihood Ratio Test, LRT, in case of random factors) and degrees of freedom (as subscript) shown. ** *p* < 0.01; *** *p* < 0.001.

		Fixed Factor	Random Factor
		Age	Site
Database	OTU richness	0.791_1,28_	2.740_1_
	Beta diversity	7.591_1,45_ **	2.001_1_
	SES-MPD	0.058_1,28_	1.372_1_
Peak profile	OTU richness	4.053_1,27_	0.000_1_
	Beta diversity	2.703_1,44_	15.445_1_ ***

## Data Availability

The data presented in this study are available in Appendix A.

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
