# Peer review of "Beta Diversity of Arbuscular Mycorrhizal Communities Increases in Time after Crop Establishment of Peruvian Sacha Inchi (Plukenetia volubilis)"

_jof, 2023, doi:10.3390/jof9020194_

Round 1
Reviewer 1 Report
Sampling design: I'm worried about the number of sampled plants. Four plants per sample is (especially when analyzing mycorrhizal fungi) very low number to compare AM communities and influences of certain parameters on it.
Line 52: why is Arbuscular Mycorrhizal Fungi written with upper caps? I suggest you use lower caps. I suggest you use abbreviation after arbuscular mycorrhizal (AM) fungi.
Line 79: use abbreviation for arbuscular mycorrhizal: AM communities. Unify across the article.
Figure 2: line 261: write y-axis
Figure 3 and 4: on second RDA axis % sign is missing, after the number.
Line 385 and onward the discussion: decide whether you will use abbreviation for chemical elements or full name. Now you have a mix of abbreviations and full names - unify throughout the discussion.
Author Response
Sampling design: I'm worried about the number of sampled plants. Four plants per sample is (especially when analyzing mycorrhizal fungi) very low number to compare AM communities and influences of certain parameters on it.
R: We have calculated species accumulation curves for each studied plot. The result showed a good coverage ranging from 100% to 88%. This information has been added to the manuscript as A3 Fig.
Line 52: why is Arbuscular Mycorrhizal Fungi written with upper caps? I suggest you use lower caps. I suggest you use abbreviation after arbuscular mycorrhizal (AM) fungi.
R: We have included the suggestion.
Line 79: use abbreviation for arbuscular mycorrhizal: AM communities. Unify across the article.
R: We have unified as AM fungal or AM fungi across the manuscript.
Figure 2: line 261: write y-axis
R: Corrected accordingly.
Figure 3 and 4: on second RDA axis % sign is missing, after the number.
R: We have corrected the figures.
Line 385 and onward the discussion: decide whether you will use abbreviation for chemical elements or full name. Now you have a mix of abbreviations and full names - unify throughout the discussion.
R: We have corrected to use abbreviations.

Reviewer 2 Report
the manuscript entitled Beta diversity of arbuscular mycorrhizal communities increases 2 in time after crop establishment of Peruvian Sacha Inchi 3 (Plukenetia volubilis), presents useful information on the continuous crop of an economic plant species from Perú. Authors molecularly characterize the AM fungal communities associated to roots in plots after different times of crop establishment, (less than one year to older than three y). 20 They analyzed the patterns of alpha, beta and phylogenetic diversity and the sources of variation of 21 AM fungal composition.
The article is well presented and analyzed; however, some details can improve the text. See below: To check for correct name: Plukenetia along the text
To highlight the economic use of the plant species and to indicate some characteristics(perennial herb? or tree?
figure Legends: to explain the variables
line 61: to indicate in the previous studies if it was a field or greenhouse study
line 88, ...and age
fig 4: if possible, to improve letter size
line343 add: also...in tropical systems
fig 3: to standardize: base sum/OR Cation sum
fig 4: MO or OM?
Author Response
The manuscript entitled Beta diversity of arbuscular mycorrhizal communities increases in time after crop establishment of Peruvian Sacha Inchi (Plukenetia volubilis), presents useful information on the continuous crop of an economic plant species from Perú. Authors molecularly characterize the AM fungal communities associated to roots in plots after different times of crop establishment, (less than one year to older than three y). They analyzed the patterns of alpha, beta and phylogenetic diversity and the sources of variation of AM fungal composition.
The article is well presented and analyzed; however, some details can improve the text. See below: To check for correct name: Plukenetia along the text
R: We have checked the Plukenetia spelling along the text, but we did not find any uncorrectness…
To highlight the economic use of the plant species and to indicate some characteristics(perennial herb? or tree?
R: We have complemented the current version by adding some characteristics and pointing out that most of its commercial use is in the pharmaceutical sector.
figure Legends: to explain the variables
R: We have rephrased the figure legends with special attention to the redundancy analysis. That has been fully explained just in Fig. 2 but not in Fig. 3 to avoid redundant information.
line 61: to indicate in the previous studies if it was a field or greenhouse study
R: We have indicated that the cited studies were arranged in greenhouse conditions.
line 88, ...and age
R: Added.
fig 4: if possible, to improve letter size.
R: We have modified the figures 3 and 4 accordingly.
line343 add: also...in tropical systems
R: added.
fig 3: to standardize: base sum/OR Cation sum
R: we have standardize to “base sum”.
fig 4: MO or OM?
R: We have changed to OM (organic matter).
